# Effect of intrapartum azithromycin on early childhood gut mycobiota development: post hoc analysis of a double-blind randomized trial

Bakary Sanyang [1], Magdalena B. Dabrowska[2,3], Nelly Amenyogbe[4], Bully Camara[1], Nathalie Beloum [1], Mariama Jammeh[1], Dodou Bojang[1], Jack Goodall [2], Nuredin Mohammed[1], Abdul Karim Sesay[1], Anna Roca [1,5,6,7] ✉ & Thushan I. de Silva [1,2,7]

Intrapartum azithromycin prophylaxis reduced maternal infections but showed no effect on neonatal sepsis and mortality. Although antibiotic exposure may indirectly alter the mycobiota (community of fungi that live in a given environment), there is no data available on how intrapartum azithromycin impacts gut mycobiota development. We hereby assess the impact of intrapartum azithromycin on gut mycobiota development from birth to the age of three years, by *ITS2* gene profiling of rectal samples from 102 healthy Gambian infants selected from a double-blind randomized placebo-controlled clinical trial (PregnAnZI-2 – ClinicalTrials.org NCT03199547). In the trial, women received 2 g oral azithromycin or placebo (1:1) during labour with the intension of assessing effect on neonatal sepsis or mortality. Secondary objectives included effects on bacterial carriage and resistance, puerperal infections, and infant growth. Our analysis show that season and parity were key factors that influenced gut mycobiota development. Intrapartum azithromycin increased the abundance of *Candida orthopsilosis* but only in the wet season and did not show different effects by sex of the child. These data suggest that season and parity can be key factors influencing gut mycobiota development and may inform strategies for a wider implementation of intrapartum azithromycin intervention.

Maternal and neonatal deaths remain leading causes of mortality worldwide, particularly in low- and middle-income countries[1]. Infections, including sepsis, contribute significantly to maternal and neonatal deaths[2,3], and projections indicate that without new interventions most low- and middle-income countries will fall short of meeting the SDG-3 targets for maternal and neonatal mortality by 2030[4,5].

Intrapartum azithromycin was developed as a potential intervention to reduce severe maternal and neonatal outcomes, including

[1]Medical Research Council Unit The Gambia at The London School of Hygiene and Tropical Medicine, Banjul, The Gambia. [2]Division of Clinical Medicine, School of Medicine and Population Health, The University of Sheffield, Sheffield, UK. [3]Florey Institute of Infection and NIHR Sheffield Biomedical Research Centre, The University of Sheffield, Sheffield, UK. [4]Department of Microbiology and Immunology, Canadian Centre for Vaccinology, Dalhousie University, Halifax, NS, Canada. [5]ISGlobal, Hospital Clínic-Universitat de Barcelona, Barcelona, Spain. [6]ICREA, Barcelona, Spain. [7]These authors contributed equally: Anna Roca, Thushan I. de Silva. ✉e-mail: anna.roca@lshtm.ac.uk

sepsis and mortality[6–8]. Two large muti-country randomized clinical trials, PregnAnZI-2 and A-PLUS, evaluated its efficacy and showed that while intrapartum azithromycin may reduce severe maternal outcomes, such as sepsis, it had no significant effects on severe neonatal outcomes, including sepsis or mortality[7,9]. However, since azithromycin reaches the baby through breast milk during at least the first 4 weeks of life[10], and antibiotics are known to influence microbiome development at very young ages[11–13], it is essential to understand the effects of the intervention on the development of the child's gut microbiome before considering large-scale implementation.

The neonatal period represents a critical window for microbiome development, with disruptions potentially causing enduring effects[13,14]. While most research on antibiotics and microbiome development have focused on bacteria, the effects on fungi or the mycobiota, have been largely overlooked. Prolonged exposure to antibiotics, especially broad-spectrum, increases the risk of fungal overgrowth. However, recent studies suggest that this is an overly simplistic view and that complex fungal-bacterial interactions are essential for a stable mycobiota[15,16]. Studies in human and murine models suggest that disruption of bacterial-fungal interactions by antibiotics could lead to imbalances affecting both communities[17,18]. For example, amoxicillin alone decreases fungal abundance while a cocktail of antibiotics with broad-spectrum activity including ampicillin, metronidazole, neomycin and vancomycin have the opposite effect[15]. Another study in humans showed that antibiotics could shift fungal interactions from mutualism to competition, with longer-lasting effects compared to bacteria[19].

Azithromycin is a broad-spectrum antibiotic with a long half-life and high tissue penetration potential. Previously, we have shown from the PregnAnZI-1[20] and PregnAnZI-2[9] trials that children whose mothers took a single dose of 2 g oral intrapartum azithromycin had short-lived alterations in both the nasopharyngeal and gut bacterial communities, with more beneficial effects in the former compared to the latter[21,22]. The effect of the intervention on the mycobiota remains unknown. The gut mycobiota plays an important role in intestinal homeostasis and pathogenesis, and its disruption has been linked to autoimmune, metabolic and neurological disorders amongst others[23].

Herein, we investigate the effect of 2 g oral intrapartum azithromycin on the development of the gut mycobiota in early childhood by ITS2 sequencing, using rectal samples collected from neonates whose mothers participated in the PregnAZI-2 trial in The Gambia. Given the limited data on mycobiota development in SSA, we also explore factors that influence infant mycobiota development in the region. Our analysis show that season and parity are key factors that shape gut mycobiota development and that intrapartum azithromycin may increase *Candida orthopsilosis* in the infant gut during the wet season.

## Results

### Summary of study participants and samples before and after processing

Baseline characteristics of the participants are shown in Table 1. The trial arms were comparable for all variables except ethnicity. All the 467 samples processed (214 azithromycin and 253 placebo) passed the NanoCLUST pipeline. A total of 133 controls including field controls ($n = 109$), extraction blanks ($n = 17$), and PCR blanks ($n = 7$) were sequenced and analyzed. After quality control filtering and removal of low-quality reads, 426 samples (197 azithromycin and 229 placebo) remained for community analysis. Details of the distribution of the remaining samples between trial arms for each time-point can be found in Supplementary Table 1.

### Within sample diversity (alpha diversity)

The results of our exploratory analysis showed that both Shannon diversity and species richness were mainly influenced by parity, season

and age (Fig. 1a, b). Both Shannon diversity and species richness increased with multiparity and wet season. Increasing age was generally associated with higher Shannon diversity and species richness, except for lower richness at the age of 3 years. Sex and some maternal ethnic groups had subtle effects on Shannon diversity and species richness, but none were statistically significant (Fig. 1a, b). As both Shannon diversity and species richness showed similar variations from the exploratory analysis, we included only Shannon diversity for the remaining alpha diversity analysis.

When comparing trial arms, intrapartum azithromycin showed varying effects on Shannon diversity with increasing age (Fig. 1c). The intervention increases Shannon diversity at day 6 and month 4 but lowers it at day 28 (Fig. 1c).

Upon further stratification to examine the effect of the intervention by season, we found that azithromycin increased Shannon diversity at day 6 during the dry season. However, the increase occurred later in the wet season, becoming only evident at month 4 (Fig. 1d). When the analysis of the impact of the intervention on Shannon diversity was stratified by parity, we observed that differences between arms occurred for primiparous mothers at day 0 and 4 months, and for multiparas from day 6 to 4 months, with a combination of increased and decreased diversity in the azithromycin arm (Fig. 1e).

Within trial arm, Shannon diversity varied with age only in the azithromycin arm with differences between day 6, month 4 and year 3 compared to day 0 (Supplementary Table 3).

### Overall community composition

Analysis of temporal variations in community composition per time-point showed no difference between trial arms except at year 3 (Fig. 2a). In both the azithromycin and placebo arms, age was a significant driver of community composition (Fig. 2b, c and Supplementary Table 4). Pairwise comparisons of day 0 against each of the subsequent time-points showed that within both trial arms community composition remained stable from day 0 up to month 4 and then changed at year 3 (Fig. 2b, c). Neither season, parity, sex, nor ethnicity showed a significant effect on community composition within both trial arms (Supplementary Table 4).

### Mycobiota taxonomic profiles and differential abundance

Both trial arms showed similar community profiles at each time-point (Fig. 3). The profiles were generally similar between day 0 and month 4, predominantly represented by *Candida* species *C. albicans* and *C. orthopsilosis*, and *Malassezia restricta* and *Zymoseptoria tritici* (Fig. 3). The year 3 samples on the other hand had different profiles compared to the previous time-points with lower abundances of *C. albicans, C. orthopsilosis*, and *Z. tritici;* and higher abundances of *M. restricta, Penicillium oxalicum*, and *Saccharomyceta* (Fig. 3). Analysis of temporal variations in taxon abundance between trial arms showed no difference at any time-point.

Upon further stratification of the community profiles of the trial arms, we observed seasonal variations (Supplementary Fig. 4a). In the wet season, abundance of *C. orthopsilosis* was relatively higher and *Aspergillus luchuensis* was lower in the azithromycin arm compared to the placebo arm (Supplementary Fig. 4b). In addition, the profiles showed relatively higher abundance of *Candida glabrata* between day 0 and day 28 in the azithromycin arm in the wet season (Supplementary Fig. 4a), though we could not test the statistical significance of this difference due to the small number of samples that remained after exclusion of low variant samples and further stratification by season.

Abundance varied significantly with age for some taxa including *Saccharomyceta* and *Cryptococcus* with the former being higher and the latter lower at year 3 compared to day 0 (Supplementary Fig. 5a, b). Abundance of *Aspergillus subgenus Circumdati* also varied with age with day 0 samples having higher abundance of this taxa compared to

**Table 1 | Baseline characteristics for the main cohort and the year 3 cohort**

| Ethnicity, n (%) | Main cohort | | | Year 3 cohort | | |
|---|---|---|---|---|---|---|
| | Azithromycin (N = 48) | Placebo (N = 54) | p value | Azithromycin (N = 42) | Placebo (N = 55) | p value |
| Mandinka | 14 (29.2) | 23 (42.6) | 0.059 | 15 (35.7) | 27 (49.1) | 0.019 |
| Wollof | 8 (16.7) | 8 (14.8) | | 8 (19.0) | 7 (12.7) | |
| Jola | 7 (14.6) | 6 (11.1) | | 7 (16.7) | 7 (12.7) | |
| Fula | 14 (29.2) | 5 (9.3) | | 9 (21.4) | 2 (3.6) | |
| Others | 5 (10.4) | 12 (22.2) | | 3 (7.1) | 12 (21.8) | |
| Maternal age (years), mean (SD) | 26.9 (5.9) | 28.6 (6.6) | 0.180 | 27.7 (6.1) | 28.5 (6.6) | 0.542 |
| Birth weight (Kg), mean (SD) | 3.1 (0.5) | 3.2 (0.5) | 0.671 | 3.1 (0.5) | 3.1 (0.5) | 0.563 |
| Sex, female (%) | 24 (50.0) | 28 (51.9) | 1.000 | 23 (54.8) | 29 (52.7) | 1.000 |
| Delivery season, wet (%)[a] | 21 (43.8) | 32 (59.3) | 0.172 | 17 (40.5) | 30 (54.5) | 0.243 |
| Age at sample collection | | | | | | |
| Day 6 (days), median (IQR) | 6 (5.5–6.5) | 6 (5.0–6.0) | | | | |
| Day 28 (days), median (IQR) | 27 (26–28) | 27 (26–28) | | | | |
| Month 4 (months), median (IQR) | 3.9 (3.9–4.0) | 4.0 (3.9–4.0) | | | | |
| Year 3 (months), median (IQR) | | | | 36.5 (34.7–39.4) | 39.4 (35.5–42.1) | |
| Mode of BF in first 6 months | | | | | | |
| Exclusive BF (%) | | | | 28 (66.7) | 37 (67.3) | 0.857 |
| Predominant BF (%) | | | | 6 (14.3) | 6 (10.9) | |
| Complementary BF (%) | | | | 8 (19.0) | 12 (21.8) | |
| Recent antibiotic consumption, n (%)[b] | | | | 6 (14.3) | 14 (25.5) | 0.252 |
| Recent sickness, n (%)[b] | | | | 18 (42.9) | 24 (43.6) | 0.664 |
| BF duration (months), median (IQR) | | | | 20 (18–24) | 18 (17–21) | 0.079 |

Sex = biological sex determined at birth. This table shows a summary of the children included in the analysis presented in this manuscript. The main cohort represents children who were initially selected for mycobiota analysis as explained in the methods. The same children were followed up at year 3, herein referred to as the year 3 cohort. Statistical tests—Chi-square test (ethnicity, sex, delivery season, mode of breastfeeding), Fisher's exact test (recent antibiotic consumption, recent sickness), two-sided independent t-test (maternal age, birthweight, breastfeeding duration).
BF breastfeeding.
[a]Wet season = June–October.
[b]Data on recent sickness and antibiotic consumption were based on 1-month recall.

each of the subsequent time-points (Supplementary Fig. 5c). We found that *C. orthopsilosis* abundance was slightly negatively correlated with maternal age (Supplementary Fig. 6).

**Mycobiota clusters**
Our unsupervised clustering analysis grouped samples into three community types, which we labeled fungCluster_1 (n = 11 out of 247; 4.5%), fungCluster_2 (n = 59; 23.9%) and fungCluster_3 (n = 177; 71.7%). Figure 4a shows the highest abundant taxa in these clusters based on hierarchical clustering. FungCluster_3 was highly represented by *Candida orthopsilosis, Malassezia restricta* and *Zymoseptoria tritici*. FungCluster_2 on the other hand had high abundance of *Candida albicans*, while the dominant taxa in fungCluster_1 were *Cryptococcus*, and *Dikarya*.

Logistic regression showed that there are trends of higher odds of having fungCluster_1 or fungCluster_2 for children in the azithromycin arm, children born in the wet season, and at the age of day 28 and month 4 (Supplementary Fig. 7). However, we found no evidence of interaction between azithromycin and season or age in driving the prevalence of these fungal community types (Supplementary Fig. 7). In addition, we did not find an association between the bacterial community types previously identified in the children and the abundance of any fungal taxa in the current analysis.

Frequencies of the clusters by time-point showed similar representation in both trial arms, with fungCluster_1 and fungCluster_2 mostly present between day 0 and month 4 (Fig. 4b). FungCluster_3 is the most frequent cluster at all time points in both the azithromycin and placebo arms, with all the children having transitioned to this cluster by year 3 (Fig. 4b). Transition of children through the different fungal clusters over time showed that children in both trial arms followed similar mycobiota transition patterns, except that transition from fungCluster_2 was relatively more stable in the azithromycin arm between day 6 and day 28 (Fig. 4c).

## Discussion
Our research contributes to the critical knowledge gap regarding gut mycobiota development in West African infants while assessing the impact of intrapartum azithromycin on this development. In the child gut mycobiota, alpha diversity was mainly influenced by age, season and parity, while community composition was stable during the first 4 months of age and the profiles predominantly represented by *Candida* species. In addition, intrapartum azithromycin had mild effects on alpha diversity of the child gut mycobiota that lasted for at least 4 months and varied again by season and parity. The intervention, however, did not have any effect on gut mycobiota community composition except at year 3.

Gut fungal alpha diversity in the normal Gambian population, as shown in the placebo arm, remained generally stable from birth to the age of 3 years but varied by season and parity. This stability in alpha diversity aligns with previous findings from children under five in Ghana and Mali[24,25]. The increase in alpha diversity during the wet season compared to the dry season likely reflects higher environmental fungal abundance due to favorable warm and humid conditions. For instance, seasonal variations in *Candida* abundance suggest environmental conditions influence fungal reservoirs, with warm temperatures and high humidity favoring growth in environmental sources[26]. The unvarying effect of sex between the trial arms imply that

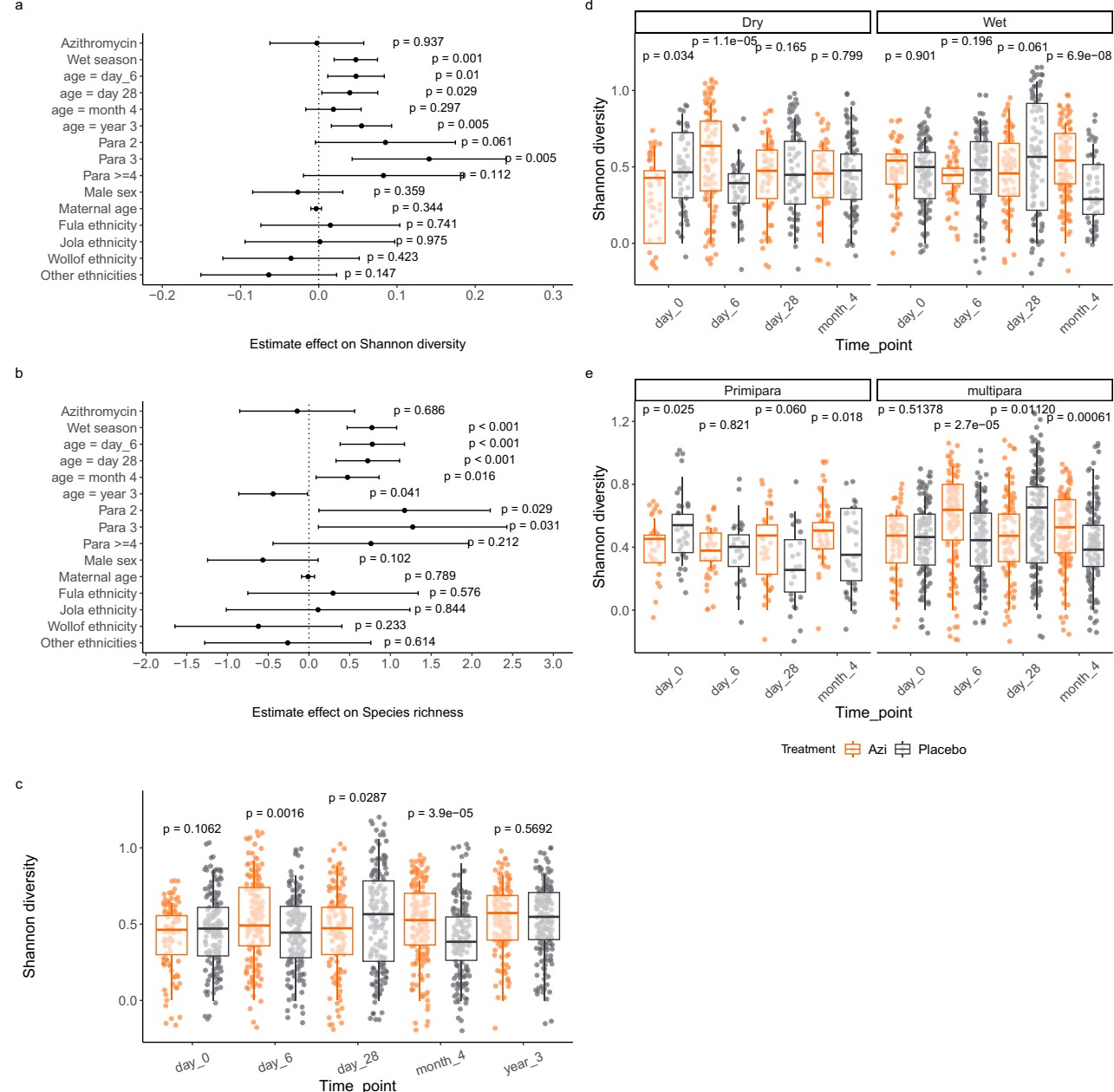

**Fig. 1 | Factors that influenced Shannon diversity and species richness.** Effect of treatment and other covariates on Shannon diversity (**a**) and species richness (**b**) estimated by multiple multivariate analysis using a linear mixed-effects model. References for comparisons: Treatment: placebo, Sampling season: dry season, Age: day_0, Parity: primipara, Ethnicity: Mandinka, Sex: female. In both figures **a** and **b**, error bars show the mean and 95% confidence intervals of standard deviation for each covariate. A total of 427 samples from 125 individuals were included. Random variations were averaged on individuals to control for repeated measurements. **c** Variation in Shannon diversity between azithromycin and placebo arms at each

time-point. **d** Variation in Shannon diversity between azithromycin and placebo arms at each time-point stratified by season. **e** Variation in Shannon diversity between azithromycin and placebo arms at each time-point stratified by parity. For all analyses of temporal variation in Shannon diversity between trial arms, unpaired Wilcoxon's Rank Sum test was used. The box and whiskers denote the distribution of Shannon diversity scores. The box shows the median and lower and upper quartiles (middle 50% of the scores), while the whiskers show the upper and lower 25% scores not excluding outliers. Para parity, Azi Azithromycin.

intrapartum azithromycin shows similar potential impact on the gut mycobiota of male and female children alike. Among children in the azithromycin arm, a rapid increase in Shannon diversity in the first week of life during the dry season is in agreement with reports of increased gut fungal alpha diversity shortly after antibiotic treatment[19,27], and may be due to the low density of *Candida* species in the gut mycobiota during this period. This is further supported by the later impact of azithromycin in the wet season where higher initial abundance of both *C. albicans* and *C. orthopsilosis* were associated with

lower Shannon diversity at day 28. Mechanisms through which antibiotics influence gut mycobiota are yet to be fully elucidated, but reports suggest possible influence through metabolic changes due to impact on the bacterial community[19,28]. In a recent study, Delavy et al.[29], showed that the impact of two types of cephalosporins on *C. albicans* abundance in the gut was inversely correlated with the level of ß-lactamase activity in the gut microbiota[29]. *Candida*, a known early gut colonizer[24,25,30], is more abundant in vaginally delivered infants[27], suggesting the maternal vaginal mycobiota as the primary source, with

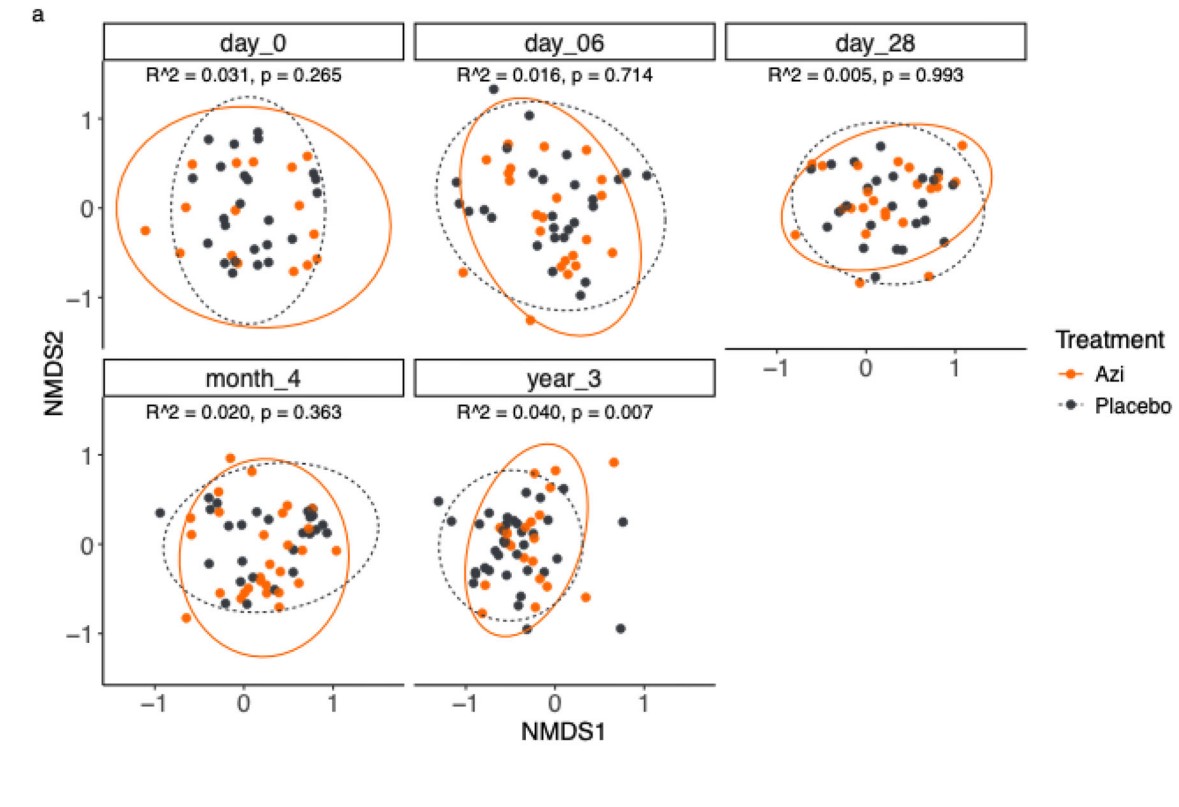

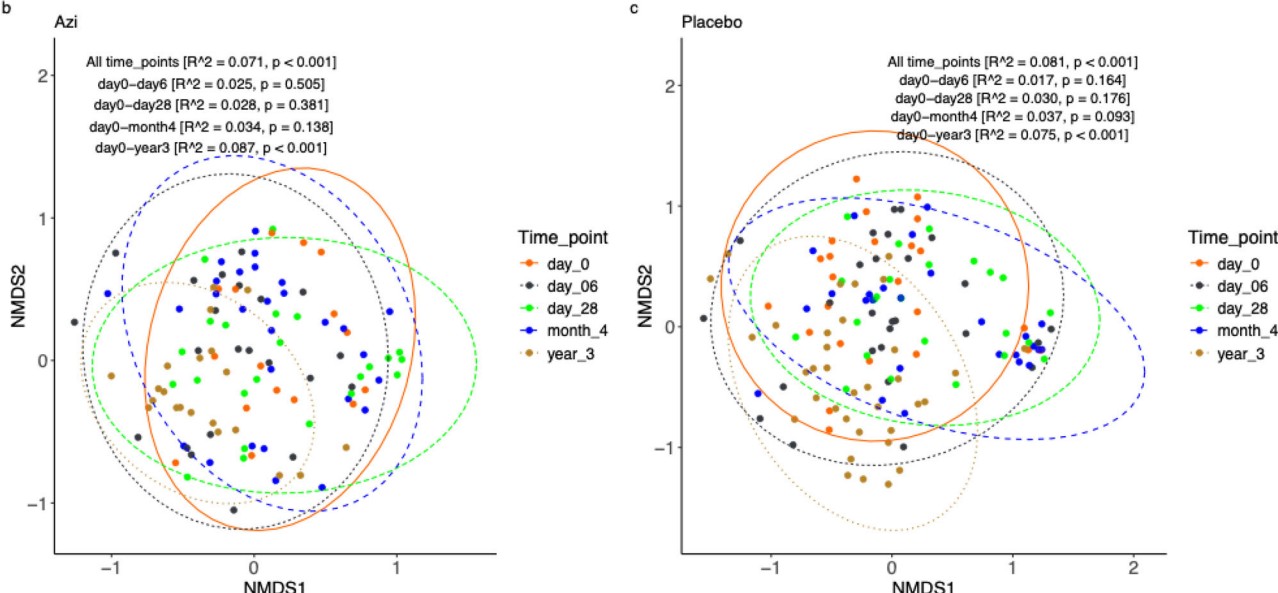

**Fig. 2 | Overall community composition. a** Community composition between trial arms at each time-point estimated by PERMANOVA. **b** Community composition by age in azithromycin arm. **c** Community composition by age in placebo arm. Age comparisons were made across all time-points with permutations restricted within individuals to control for repeated measurement, and pairwise between day 0 and each of the subsequent time-points. Azi Azithromycin.

environmental contributions playing a secondary role. The inverse correlation of abundance of *C. orthopsilosis* with maternal age suggests that babies of young mothers are likely to carry more of this fungi compared to other children, with potential further promotion by azithromycin. As high abundance of *Candida* in the gut is associated with gut inflammation[23], this suggests that the risk of *Candida* infections in neonates born to young mothers who receive intrapartum azithromycin needs to be further investigated.

The increase of fungal alpha diversity by parity is notable, though not previously reported. This parity-associated alpha diversity increase may stem from changes in the maternal vaginal microbiota, where bacterial diversity increases during the peripartum period. Specifically, it has previously been reported from a study in USA that the vaginal microbiota changes dramatically after delivery with expansion of proinflammatory bacteria that could last up to 18 months[31], potentially influencing fungal transfer to infants. Therefore, birth spacing which affects vaginal microbiota recovery, may play a role, though this was not assessed in our study.

Intrapartum azithromycin did not have any effect on community composition during the first 4 months of life in our study. Our findings

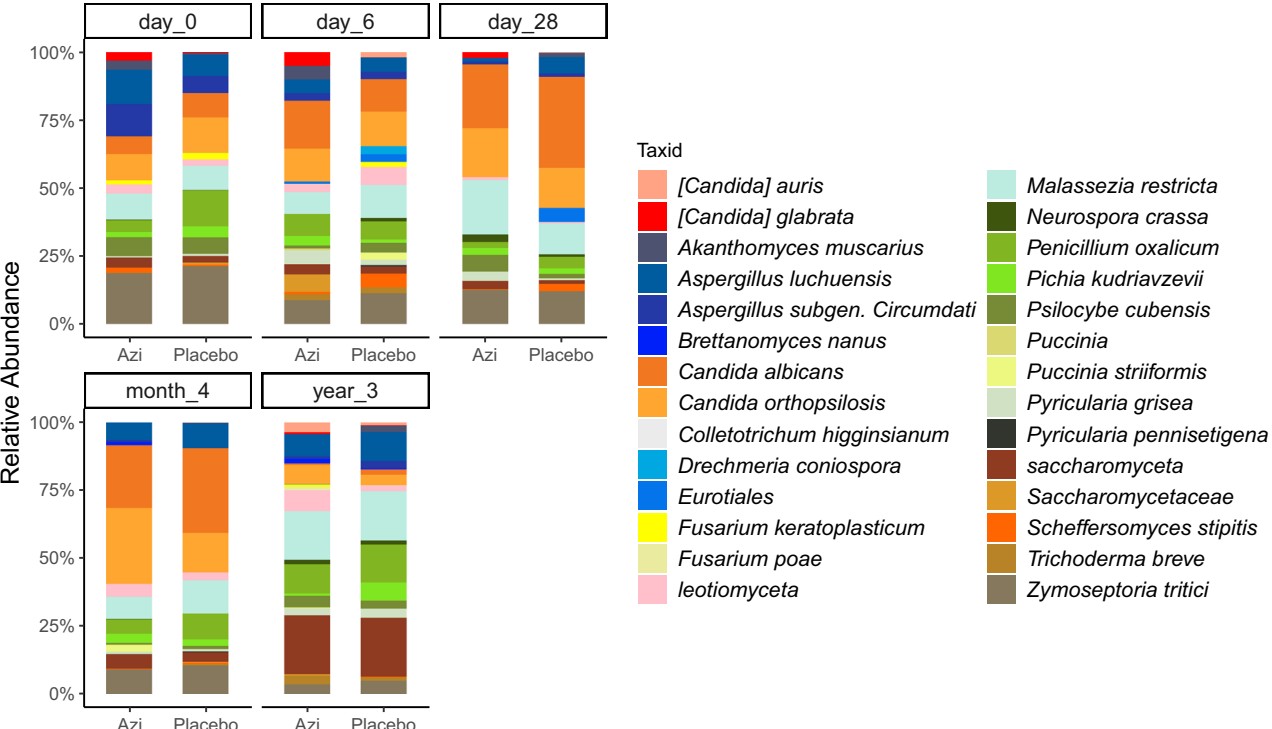

**Fig. 3 | Gut mycobiota community profiles by treatment over time.** Community profiles showing relative abundance of the top 28 fungal taxa between azithromycin and placebo arms at each time-point. Azi Azithromycin.

are supported by a report by Seelbinder et al.[19], which showed that azithromycin had a mild impact on gut mycobiota composition compared to other antibiotics such as amoxicillin-clavulanic acid and ciprofloxacin. Therefore, we cannot explain the difference observed in community composition between the trial arms at year 3. However, given the time interval between month 4 and year 3, it is possible that this difference is due to other factors that we were not able to control for. We therefore recommend similar assessment in future studies to verify this effect.

The overall stability of gut fungal community composition in both trial arms during the first 4 months of life reflects the high degree of homogeneity of the taxonomic profiles during this period. Neither season nor parity significantly influenced community composition, implying that age was the primary driver. This age-related change is likely influenced by diet[23], as diet had already changed for all the study children by year 3. This is consistent with the observed increase in *Saccharomyceta* at year 3, aligning with similar findings by Schei and colleagues[32].

Children in both trial arms followed similar mycobiota development trajectories. FungCluster_3, the dominant cluster, represents a diverse and balanced community type typically represented by species commonly found in similarly aged children in West Africa[24,25]. Its high prevalence at all time-points suggests that it reflects the normal mycobiota community type for the study population. In contrast, fungCluster_1 and fungCluster_2, found up to month 4, exhibit characteristics of a dysbiotic mycobiota, with high abundances of opportunistic pathogens including *Cryptococcus* and *C. albicans*, respectively. Although evaluating the association between fungal community types with the risk of disease was beyond the objective of the current study, higher prevalence of fungCluster_1 and fungCluster_2 at day 28 and month 4, suggests a potential increased risk of *Candida* and *Cryptococcal* infections beyond the neonatal period. These two clusters showed a weak association with azithromycin during the first 4 months of life, suggesting a potential increase among children exposed to the intervention. Supporting

this observation, Drummond et al.[17], reported in a mouse model that exposure to wide-spectrum antibiotics impairs gut mucosal immunity by disrupting the production of IL-17A and granulocyte-macrophage colony-stimulating factor (GM-CSF) from CD4 T-cells and innate lymphoid cells in the gut[17]. While human studies are needed to confirm those findings, it implies that early acquisition of fungCluster_2 may increase the risk of invasive candidiasis whilst azithromycin still reaches the child[10]. Further supporting this is the association between *Bifidobacterium* and immune development. Findings from the study of a Finnish infant cohort revealed that immune dysregulation in early life could be a consequence of Bifidobacterial depletion[33], and we have shown previously that azithromycin decreases Bifidobacterial abundance in the first 4 weeks of life[22]. In a USA infant cohort, high abundance of *Candida* coupled with low Bifidobacterial abundance has been associated with increased risk of atopy and asthma[34]. Immunological surveys are therefore necessary for a more comprehensive understanding of the effects of the mycobiota and microbiota alterations observed in our study children on immunity.

The major strength of this study is the randomized clinical trial design which allows for the evaluation of the impact of intrapartum azithromycin on the development of the child gut mycobiota. However, there are some limitations that also need to be highlighted. First, we used RS samples from a previous study we conducted to investigate the effect of intrapartum azithromycin on gut microbiota development. As a result, the sample size was not specifically calculated for mycobiota analysis. Given the high variability in the gut mycobiota compared to the microbiota, some weak associations observed between azithromycin and the mycobiota could be due to insufficient statistical power. Therefore, larger studies will be necessary to confirm some of our findings. Second, the study included only healthy children with no cases of sepsis or severe infection, limiting the ability to directly link mycobiota changes to clinical outcomes. Third, the participants in this study were recruited from peri-urban settlements in The Gambia and

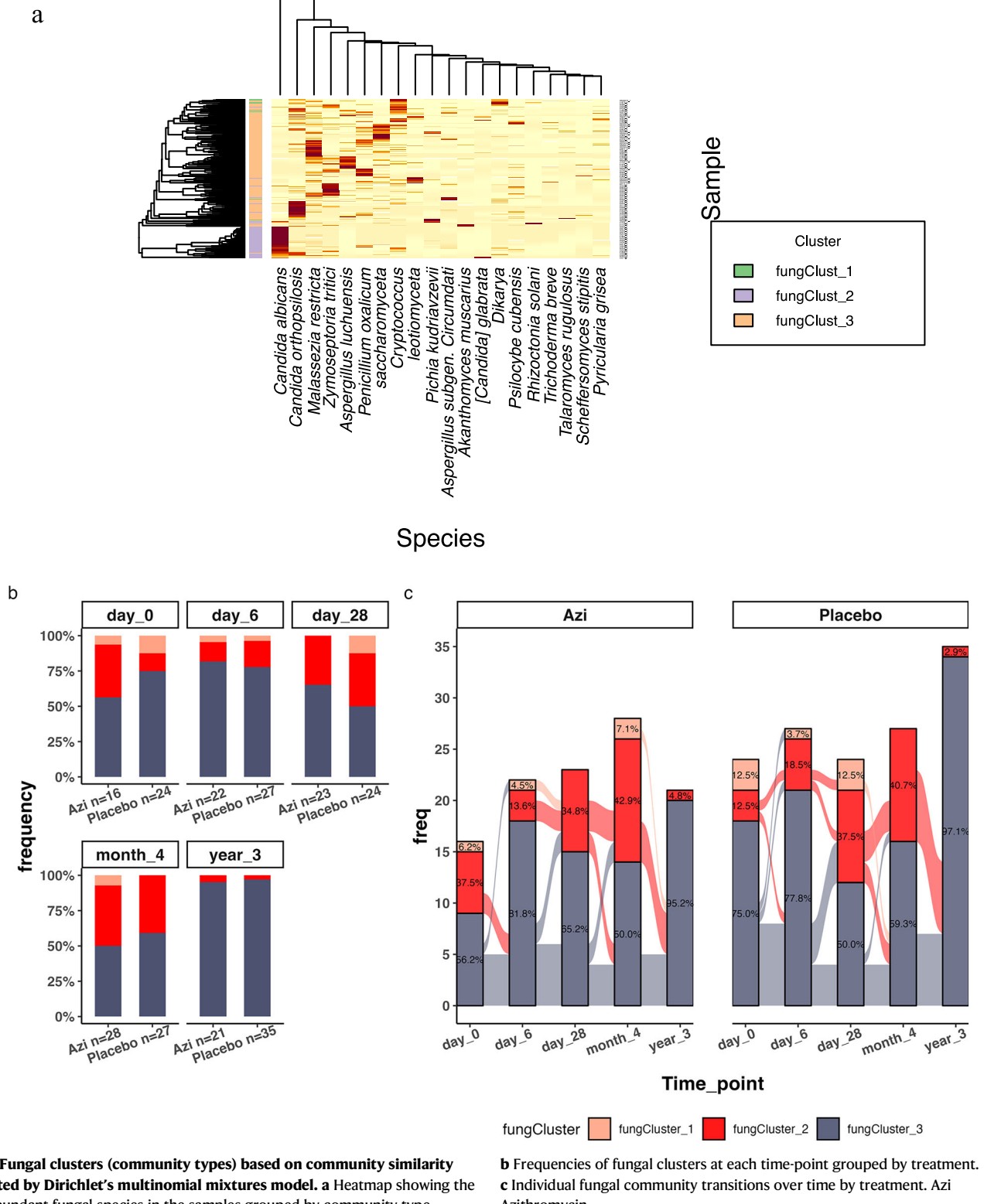

**Fig. 4 | Fungal clusters (community types) based on community similarity generated by Dirichlet's multinomial mixtures model. a** Heatmap showing the most abundant fungal species in the samples grouped by community type. **b** Frequencies of fungal clusters at each time-point grouped by treatment. **c** Individual fungal community transitions over time by treatment. Azi Azithromycin.

therefore our results may not be entirely generalizable to typical rural sub-Saharan African settlements. Fourth, our study presents compositional data (relative abundance) and therefore does not give information on the absolute quantity of fungi in the gut of the children. However, relative abundance is a reliable measurement as it remains unaffected by total abundance which can vary between samples due to uneven sequencing depth or sample biomass.

Finally, potential contamination from the sample storage medium (STGG) could have influenced the results, particularly *Saccharomyces cerevisiae*. While *Saccharomyces* has been reported as a normal component of the infant gut mycobiota[24], the exceedingly high abundance of *S. cerevisiae* in our STGG media, likely due to its use in skimmed milk to control aflatoxins[35], required its removal from the dataset.

Our study shows similar gut mycobiota development pattern as what has previously been reported by other studies from West Africa, exhibiting high *Candida* abundance in the first few months and a stable alpha diversity over the first 3 years of life. In addition, our study also revealed a significant influence of season and parity on the gut mycobiota, with birth during the wet season being associated with increased *Candida* abundance and multiparity with increased fungal alpha diversity. Overall, intrapartum azithromycin showed mild effects on gut mycobiota development, which did not vary by sex of the child. The observed seasonal and parity-specific variations in its impact highlight the complexity of these effects. The increase of *C. orthopsilosis* observed in the first 4 months of life in the azithromycin arm during the wet season is an important finding for implementation of this intervention. Combined with the potential for antibiotic-induced gut barrier impairment[17], this raises - questions about the potential for increased risk of *Candida* infections among children exposed to this intervention during the wet season. Moreover, the association between young maternal age and increased abundance of *C. orthopsilosis* indicates a potentially higher vulnerability in infants of young mothers. This understanding on how maternal factors interact with azithromycin exposure to influence gut mycobiota and infection risk is vital for tailoring the intervention. Given the known variations in gut mycobiota dynamics across populations, it is essential to investigate the impact of intrapartum azithromycin in diverse populations. Such data is crucial for informed decision-making about large-scale implementation of the intervention in regions where maternal and neonatal health burdens are highest.

## Methods

### Ethical approval
The main trial (PregnAnZI-2) was approved by the Gambia Government-MRCG Joint Ethics Committee, the Comité d'Ethique pour la Recherche en Santé, the Ministry of Health of Burkina Faso, and the London School of Hygiene and Tropical Medicine Ethics Committee. In addition, this post hoc study was approved by the Gambia Government-MRCG Joint Ethics Committee and the London School of Hygiene and Tropical Medicine Ethics Committee. Study women signed informed consent for the trial during their antenatal visits and samples collected during the 4-months follow-up. An additional consent was sought from mothers of the study children for the 3-year survey.

### Study design
PregnAnZI-2 (ClinicalTrials.org NCT03199547) was a phase-III, double-blind, randomized, placebo-controlled trial in which 11,983 women from The Gambia and Burkina Faso were randomized to receive a single dose of 2 g of oral azithromycin or placebo (ratio 1:1) during labor. The median time from administration of azithromycin to delivery was 1.6 h (IQR 0.5–4.1)[9]. Details of inclusion and exclusion criteria are available in the study protocol published elsewhere[36]. The primary objective of the trial was to evaluate the impact of the intervention on neonatal sepsis and death[9]. The trial showed no effect on the neonatal sepsis or death though other infections, including skin infections, were reduced in the azithromycin arm[9]. Also, the intervention reduced maternal infections including mastitis[9].

Here only participants from sites in The Gambia were included, where the trial was conducted in two health facilities located in the coastal region of the country. A subset of 253 mother-baby pairs were recruited in a bacterial carriage sub-study. These were randomly selected from study children born between January 2019 and March 2020[36]. All the children in the carriage sub-study were vaginally delivered.

### Selection of children for mycobiota analysis
The first 102 children recruited into the bacterial carriage sub-study in the Gambia were selected. For these children, an additional follow up was conducted at the age of 3 years, during which rectal swabs were collected from 72 of the 102 children initially selected plus an additional 25 children from the main carriage cohort. Details of the sample selection process is illustrated in Fig. 5. A total of 126 children were included in this mycobiota study. Fifty-seven of these children were born to mothers who received the active intervention (2 g azithromycin) and 69 were born to mothers who received placebo.

### Collection of rectal swabs
During the trial, rectal swabs (RS) were collected with FLOQSwabs (COPAN, REF:519CS01) from the children included in the bacterial carriage sub-study. The first swabs were collected shortly after delivery (day 0). After post-delivery hospital discharge, active-follow up with home-visits occurred at day 6, day 28 and 4 months. RS were collected from the children during these follow-up visits. All the swabs were stored in skim milk-tryptone-glucose-glycerin (STGG) transport medium without preservative and transported in temperature-monitored (2–8 °C) cooler boxes with ice packs to the laboratory within 8 h. Upon reception, the samples were homogenized and stored at −80 °C for later processing.

### Rectal swabs used for mycobiota analysis
A total of 467 RS from the children (214 azithromycin, 253 placebo) were processed. Sample numbers per time-point ranged from 40 to 46 for azithromycin arm and from 48 to 53 for placebo arm (Supplementary Table 1). As per best practice for microbiome studies, field controls were collected during the main study (PregnAnZI-2 trial) and during the year 3 follow up of children included in the microbiome studies. A vial of STGG with a plain sample collection swab immersed and exposed for approximately 1 min to the environment where participants were sampled, was used as a field control. To avoid contamination with fecal matter from the samples, the field controls were collected before the rectal swabs. A total of 109 field controls were collected, of which 12 were collected at the hospital sites during the main trial follow ups and 97 were collected from individual participant households during the year 3 follow up.

### DNA extraction and quality control
We extracted DNA from the RS using the DNeasy® PowerLyzer® PowerSoil® Kit from Qiagen (Qiagen, Germany) with slight modifications as follows. A total of 100 μl of homogenized RS in 1 ml of STGG transport medium was used as input and two rounds of 10 cycles of beat beating at 2500 RPM for 30 s was applied with 10 min break between rounds. We followed the manufacturers protocol for the rest of the extraction steps and eluted in 100 μl of elusion buffer from Qiagen. Blank extraction and field controls were included and taken through all downstream analyses. Plain sterile swabs in sample storage vials with STGG exposed to the environment at the field site were included as field controls. The DNA extracts were quantified with a Qubit 4·0 Fluorometer (Invitrogen/Thermo Fisher Scientific) using the double stranded DNA high-sensitive kit.

### ITS2 library preparation and sequencing
ITS2 amplicons were generated using the following primers (forward: GTGAATCATCGARTCTTTGAAC, reverse: TATGCTTAAGTTCAGCGGGTA) published elsewhere[37], and the Q5 Hi-Fi 2X master mix from New England Biolabs (NEB). The PCR reaction was set up with 12.5 μl of master mix, 1 μl each of forward and reverse primers at 10 μM concentration, 5 μl of template, and 5.5 μl of molecular grade water. The following thermocycling conditions were used. Initial denaturation at 98 °C for 30 s, followed by 35 cycles of denaturation at 98 °C for 20 s,

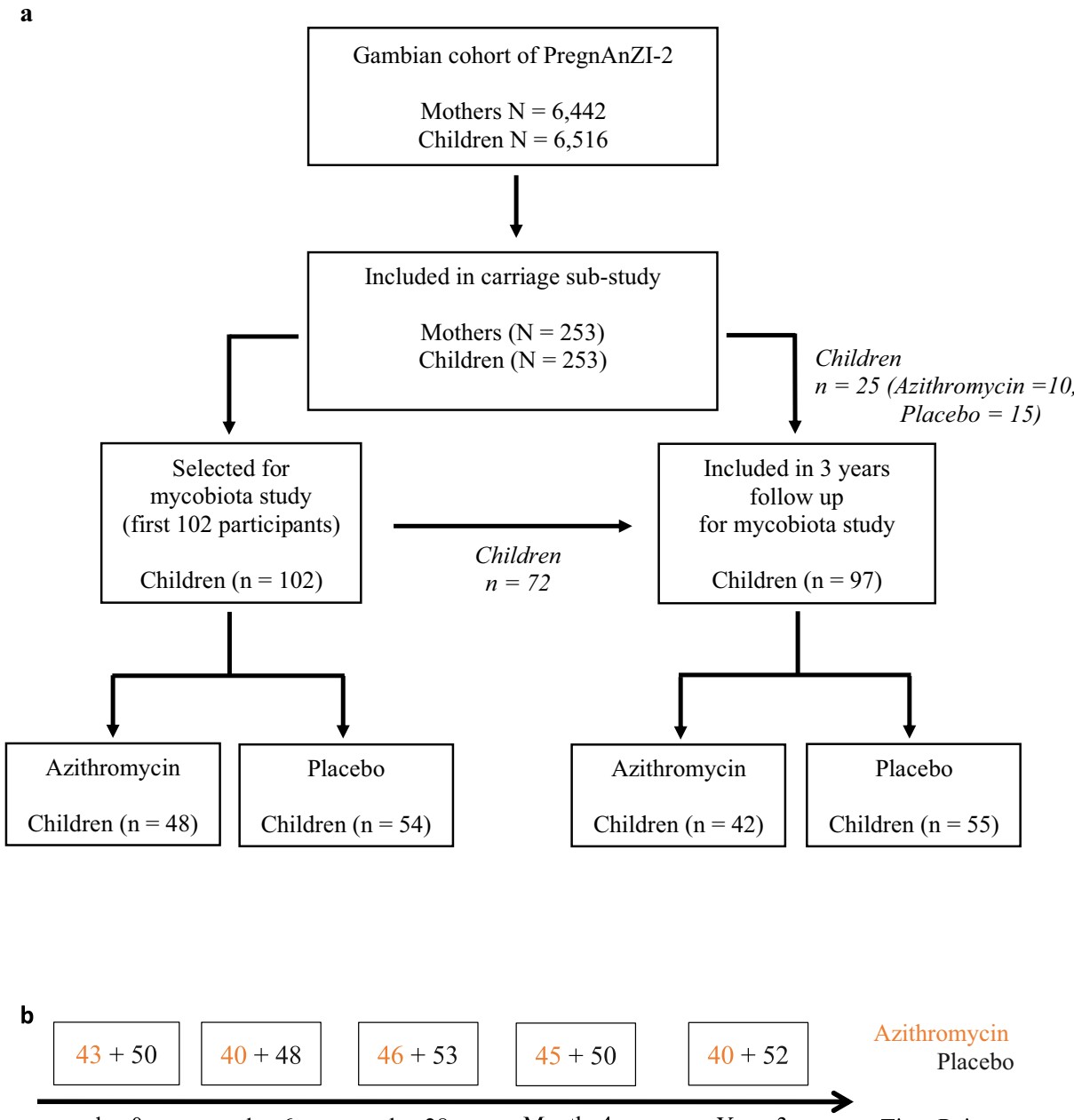

**Fig. 5 | Study profile summarizing sample selection process. a** Sample selection for mycobiota study. **b** Distribution of samples included in mycobiota analysis between trial arms at each time-point.

annealing at 51 °C for 30 s, and extension at 72 °C for 20 s. This was followed by a final extension at 72 °C for 10 min. The expected amplicon size was ~350 bp. Post PCR DNA concentrations were quantified as described earlier. The PCR products where then normalized to 50 ng (~200 fmol) in 12.5 µl for library preparation. Nanopore libraries were made using the ligation sequencing kit (SQK-LSK109) with the native barcoding expansion 96 (EXP-NBD196) using manufacturers protocol. Six pools of 96 libraries and 1 pool of 39 libraries were generated including rectal samples, field controls, extraction reagent blanks and PCR reagent blanks. Each pool was sequenced using an R9 MinION flow cell (FLO-MIN106D) on a GridION using quality filters of 200 bp minimum read length and minimum quality score of Q9. Between 4 million to 8 million reads were generated for each of the 96 library pools, and about 400,000 reads were generated for the pool of 39 libraries, which were mostly field controls.

### Analysis of ITS2 sequences

The sequence data were analyzed using NanoCLUST[38], a pipeline designed for analyzing 16S rRNA gene amplicon data generated on the Oxford Nanopore Technology (ONT) sequencing platform. The pipeline was optimized for analyzing ITS2 data generated on the ONT sequencing platform. The pipeline takes in raw FASTQ files and performs trimming and filtering using fastp[39]. Reads were filtered by minimum $Q$-score of 9 and read lengths between 200 bp and 500 bp. It then generates k-mer frequencies and performs clustering using UMAP-HDBSCAN[40,41]. We subsampled up to 5000 reads per sample for clustering setting minimum cluster size at 50 and minimum distance to define a cluster at 0.1. Sequences in each cluster are then corrected using Canu assembler[42]. This is followed by the selection of draft representative sequences for each cluster, which are subsequently polished using Racon[43], and Medaka[44]. The polished consensus

sequences are then classified using Kraken2[45], with a subset of the standard Kraken2 database (April 2024 version) containing only fungal and human sequences.

## Mycobiota community analysis

Classification data generated with NanoCLUST were imported into R (version 4.4.0) and RStudio (version 2024.04.1 + 748)[46], for further microbial community analysis. The Tidyverse (version 2.0.0), and phyloseq (version 1.48.0)[47], packages were mainly used for data handling and organization. First, taxon counts were normalized by total sum scaling. We then inspected the initial profiles of the samples and controls to see the relative abundances of human and unclassified reads as well as possible contaminants (Supplementary Fig. 1). Most of the reads in the PCR and extraction blanks were unclassified, and the top two taxa in the rectal swabs were unclassified and human (Supplementary Fig. 1). We filtered out all human and unclassified reads. The filtered dataset was transformed into a feature table and used to create a phyloseq object together with the sample metadata. Using decontam (version 1.20.0)[48], we identified contaminants in the dataset using the isContaminant function in the combined mode setting a probability cutoff of 0.2 for calling a contaminant. A total of 18 taxa (Supplementary Table 2) were identified as contaminants and filtered from the dataset. We re-inspected the profiles of the samples and controls again to see how they have been impacted by the filtering. Most of the potential contaminants were removed, though there still remained a couple of other potential contaminants (*Aspergillus fumigatus* and *Aspergillus puulaauensis*) which were subsequently filtered out (Supplementary Fig. 2). Details on the sample numbers that were available before and after all the filters were applied are shown in Supplementary Table 1. The number of reads in the samples and controls before and after filtering of unclassified, human, and contaminants are shown in Supplementary Fig. 3. Samples with only 1 or 2 species (not read counts) remaining post-filter were excluded from analyses of community composition, clustering, and differential abundance to avoid outlier effects.

Before measuring the effects of the intervention on the mycobiota, we first assessed the trial arms for comparability at baseline. Variables that had different representation between the trial arms were controlled for during statistical analysis.

We measured diversity within individual samples (alpha diversity) using Shannon index and species richness. To understand the effects of potential factors that influence microbiome diversity, we assessed the effects of treatment, ethnicity, season of sampling, parity, and sex (sex was determined at birth) among others as covariates on Shannon diversity and species richness using a linear mixed effects model with a random effect on subject. Based on this model we examined interaction between treatment and season measuring variation of Shannon diversity between trial arms at each time-point by season using Wilcoxon rank sum test. The season stratified analysis did not include samples collected at the age of 3 years because all the samples at this time point were collected during the wet season. We also examined interaction between treatment and parity in a similar analysis. Finally, we assessed the effect of age on Shannon diversity using a linear mixed effects model, comparing day 0 to each of the subsequent time-points while controlling for treatment, season of sampling, ethnicity, parity and sex and adding individual variations as random effects. *p* values were adjusted for repeated testing by Dunnett's test.

Overall community composition was measured with Bray–Curtis index using the vegdist function in vegan (version 2.6.4)[49]. Variance in community composition was compared between groups by permutational multivariate analysis of variance (PERMANOVA) using the adonis function in vegan. We assessed the effects of age, season of sampling, ethnicity, parity, and sex on overall community composition

within each trial arm. For age comparisons, we compared variance across all time-points and then between day 0 and each of the subsequent time-points. We then compared variance in community composition by treatment by comparing trial arms at each time-point. As we did not find any significant variation in community composition by sampling season, parity, ethnicity, or sex, no stratified analysis was done for these variables.

To further explore the structure of the gut fungal community in our cohort, we employed an unsupervised clustering using Dirichlet's Multinomial Mixtures model implemented in mothur (version 1.44.0)[50], to group samples into clusters based on their community types. We used logistic regression to examine association between fungal clusters (grouped as dominant community type which we called fungCluster_3 vs others) and covariates including treatment, age, sampling season, and parity. We visualized the profiles of the clusters using a heatmap to see which taxa were driving the separation of samples and summarized the frequencies of the clusters per time-point for each trial arm. To understand the dynamics of changes in fungal community types, we assessed the transition of individual children through the different clusters over time.

We assessed variations in taxon abundance using linear models implemented in MaAsLin2 (version 1.18.0)[51]. First, we assessed temporal variations in taxon abundance between the trial arms at each time-point. We then assessed differences in taxon abundance between trial arms by season of sampling. Finally, we assessed overall abundance for each taxon by treatment, age, season of sampling, parity, ethnicity, sex, and bacterial cluster, while adjusting for random effects on subject. Only taxa with minimum abundance and variance of at least 0.01% were included in differential abundance analysis. *p* values were adjusted for multiple testing and an FDR cutoff of 0.2 was used for calling significance. We used bacterial clusters identified in these children from our previous study[22], where we characterized gut microbiota development to identify potential associations of the bacterial community types with the abundance of individual fungal taxa. Bacterial clusters identified in the children at day 28 were used as this was the point at which we saw marked differences in the representation of bacterial community types between the trial arms[22]. Data on the bacterial clusters assigned to individual samples is provided in Supplementary Data 1[52].

## Statistics and reproducibility

Sample size for each trial arm at baseline is shown in Table 1. Demographic comparability of the trial arms at baseline was assessed using various tests. For categorical variables (ethnicity, sex, season, breastfeeding mode) except for recent sickness and recent antibiotic consumption, Chi-square test was applied. For recent sickness and recent antibiotic consumption, Fisher's exact test was applied. For continuous data (maternal age, birthweight, breastfeeding duration), *t*-test was applied.

The sample size was originally calculated for gut microbiota analysis. We chose to work with the same set of samples because we intended to link our findings with the results we obtained from the gut microbiota analysis. The sample size calculation was based on power to detect at least 10% difference in the top 10 operational taxonomic units (OTUs) and 20% difference in the next 10 OTUs in the gut microbiota using a sample size and power calculation tool for case-control microbiome study design developed by Mattiello et al. [53]. We used the gut microbiome dataset from the human microbiome study embedded in the tool. With a sample size range of 30–70 per group, we estimated power by Monte Carlo simulations with 100 replications using the top 50 OTUs from the dataset. A sample size of 45 per group at each time-point had over 90% power to detect these differences. Though we expect higher interpersonal variations with the gut mycobiota than the gut microbiota,

our analysis factored this by exclusion of outliers in comparisons of community composition and differential taxon abundance. The remaining samples used in each of these analyses are shown in the respective results.

Multivariate analysis of factors that influenced alpha diversity was done using a linear mixed effects model with random effects on individuals. Temporal variations in alpha-diversity between trial arms was assessed at each time-point by Wilcoxon Rank Sum test. Changes in Shannon diversity over time was assessed using a linear mixed effects model including ethnicity, season, parity, and sex as covariates in addition to the trial treatment while controlling for random variations among individuals. *p* values were adjusted by Dunnett's test.

Beta-diversity was compared between trial arms at each time-point by permutational multivariate analysis of variance (PERMA-NOVA) using the adonis function in vegan[49]. Beta-diversity was also compared by age across timepoints using the adonis function with permutations restricted within individuals.

Differential taxon abundance was assessed using generalized linear models implemented in MaAsLin2[51]. Temporal differences between trial arms were assessed at each time. Variations by age and other covariates were assessed with the aggregated data.

Analysis of factors that drive mycobiota community-types was done by logistic regression using a generalized linear model.

### Reporting summary

Further information on research design is available in the Nature Portfolio Reporting Summary linked to this article.

## Data availability

The sequence data generated in this study has been deposited in the SRA under the bioproject accession PRJNA1129542. The data on bacterial community types of the samples used in this analysis is available on figshare at this link https://doi.org/10.6084/m9.figshare.29357054.52 Additional demographic data on the study participants is available upon request. The data in this study has been collected following provision of informed consent under the prerequisite of strict participant confidentiality. Access can be requested through the Gambia Government/MRC Joint Ethics Committee. The review process and release of data will be facilitated by MRC Unit The Gambia (http://www.mrc.gm/) through the Head of Governance Mr Dembo Kanteh (Dembo.Kanteh@lshtm.ac.uk) and the corresponding author A.R. (aroca@mrc.gm). The scientific merit of the request will be evaluated by the Scientific Coordinating Committee at MRCG at LSHTM. Response would be given within 8 weeks. All other data are available within the Article and Supplementary files.

## Code availability

The pipeline used for analysis of the raw sequences is available at https://github.com/Clinical-Infection-Research-UoSheffield/nf-core-nanopath. The codes used for mycobiota community analysis in R are available on GitHub at this link https://github.com/Baksso/Azimic/blob/main/MIC2_gut_mycobiota_analysis.Rmd, and also available on Zenodo[54].

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

## Acknowledgements

We would like to extend our sincere appreciation to the PregnAnZI-2 study participants, both mothers and their children, for taking part in this trial. We would also like to acknowledge the entire PregnanAnZI-2 team, including Prof Umberto D'Alessandro and Prof Halidou Tinto (co-investigators), the coordination team in the Gambia (led by Dr Bully Camara), the field, lab, and the data management teams. We thank the funders of this study UK Research and Innovation (MC_EX_MR/P006949/1) received by A.R., Bill & Melinda Gates Foundation (OPP1196513) received by A.R., and MRCG@LSHTM Doctoral Training Program received by B.S. We also thank our collaborators who have contributed to this work, including the Ministry of Health, Gambia, Bundung Maternal and Child Hospital, Serrekunda Health Centre, the MRCG@LSHTM Genomics Core Platform, and the Clinical and Infection Research Group at the University of Sheffield. Finally, we would like to thank CLIMB-BIG-DATA for giving us access to their cloud computing resources which we used for part of our analysis.

## Author contributions

This study was conceived and designed by A.R. supported by A.K.S., T.d.S and B.S. Sample selection was done by A.R. and B.S. The field work was led by B.C. and N.B. The laboratory work including DNA extraction, library preparation, and sequencing was carried out by B.S. with support from M.J. and D.B., supervised by A.K.S and T.d.S. ITS2 sequence data analysis was carried out by B.S. with support from M.D. and J.G. Mycobiota community analysis was carried out by B.S. under the supervision of T.d.S. and A.R. supported by N.A. N.M. contributed to the statistical analysis. B.S., A.R. and T.d.S. drafted the manuscript. N.A., A.K.S., J.G., B.C., N.B. and N.M. contributed to editing the manuscript. All authors approved the last version of the manuscript.

## Competing interests

The authors declare no competing interests.
