## [Transparent Peer Review file · Nature Communications]

Effect of intrapartum azithromycin on early childhood gut mycobiota development: post hoc analysis of a double-blind randomized trial

Corresponding Author: Professor Anna Roca

Version 0:

Reviewer comments:

Reviewer #1

(Remarks to the Author)

This is a well-written and solid manuscript. The importance of the study is well defined and leaves no question. The fungal gut microbiota is often overlooked and might have unexpected effects on particularly the infant health outcomes. This study both highlights the infant gut mycobiota development in the beginning of life, as well as the impact of azithromycin on the gut mycobiota composition.

There are only minor comments needing clarifications:

1. What was the correct number of samples in the analyses? In the Table it is written that main cohort includes n=48 in the azithromycin group and n=54 in the placebo arm. However, in the abstract the numbers of selected children are n=57 and n=70. Use in the abstract the numbers corresponding to the main cohort (and main analyses).
2. In the results the authors have nicely shown how different background factors associated with diversity and richness. Are these background factors included as confounding factors in the statistical tests? Or is the cohort always stratified according to these in the tests? Maybe it could be included as a confounding factors to the tests in some cases to see if the outcome is the same?
3. In Fig 2 the diversities are difficult to read when the background lines divide the clusters (it is difficult to see which were the actual comparisons). Consider moving the lines so they are supporting the wanted comparison.
4. Were all children delivered vaginally?
5. In Table 1 recent antibiotic use was presented. Did that effect the mycobiota?
6. Table 1 is divided into two parts. When looking just at the table it is not clear whether the study consist of two completely different cohorts at two different timepoints or a single cohort that was followed up to 3 yrs. In Fig 1b the number of samples in control and AB treated group is outlined. From there it seems to be one cohort following the same patients. Please clarify this in the legend.
7. Discuss the limitation of using relative abundances (instead of including quantitative data as well).
8. References are not in a uniform format. Please revise.

Reviewer #2

(Remarks to the Author)

Reviewer #3

(Remarks to the Author)

Thank you for an insightful manuscript. Below, I have given an overall review of the manuscript by topic, and after that, some specific suggestions to help the authors revise the manuscript. With this review, I recommend that this manuscript will go back to the authors for revisions before further consideration for publication.

Key results

Reviewer: The study explores the effects of antibiotics on the understudied mycobiota of the human gut. Furthermore, the trial took place in Africa, which is generally underrepresented in microbiota studies. The sample size was quite large, probably making it one of the largest longitudinal studies of its kind. The use and utilization of a large negative control sample group was impressive and left me confident that the findings largely represent a true human mycobiota rather than contamination.

Validity of the manuscript

Reviewer: I do not see immediate flaws that would prevent the publication of the manuscript outright. Some issues require addressing by the authors, especially regarding the quality of the figures, as some of the figures were so unclear it was difficult to verify the results. With sufficient revisions the manuscript could qualify for publication.

Originality and significance

Reviewer: The conclusions are original. This was a large trial with an underrepresented study population in microbiome research. I think the results should be interesting to those in the field, especially those interested in the fungal side of the microbiome, but also those interested in the effects of antibiotics on the gut microbiome.

Data & methodology

Reviewer: The quality of the data seems to be good. Methods seem appropriate for the study question. A very large negative control sample group was included, and it seemed to have been utilized well in identifying contaminant reads from the samples. The reporting of the data and methods seems sufficient for reproduction of the results.

Appropriate use of statistics and treatment of uncertainties

Reviewer: Error bars are not defined in the corresponding figure legends. This needs to be corrected with revisions. Otherwise, the statistics are generally sound. Minor clarifications are needed, as addressed in the Reviewer questions and suggestions below.

Conclusions and data interpretation

Reviewer: I think the data has been well interpreted although some of the figures need enhancing for me to be sure. The conclusions are fine, although I remain doubtful of whether the observed effect of multiparity is rather due to the age of the mother or some other undiscussed effect taking place in the study.

Suggested improvements/additional data

Reviewer: I want to highlight my suggestion from the Reviewer questions and suggestions regarding the data availability statement. Raw data should be uploaded to a public repository, and if this has been done already, an identifier to the data should be provided.

Reference list

Reviewer: Overall, the reference list seems good. Further discussion, and therefore inclusion of additional related references, could be provided regarding how antibiotics affect the mycobiota composition, considering antibiotics do not directly target fungi.

Clarity and context

Reviewer: The abstract is clear enough. The introduction is overall appropriate, although a bit long in my opinion. As the authors state in the limitations, the study doesn't allow straightforward conclusions on any clinical outcomes, which is why especially the beginning part in the introduction regarding maternal and neonatal deaths and severe outcomes seems a bit out of place. With this in mind, I think the focus here should rather be the mycobiota composition and possible effects on it. Speculating the effect of disrupted mycobiota on clinical outcomes would be fine in the discussion. The conclusions seem overall fine and reflect the results in this manuscript.

Reviewer questions and suggestions:

-Methods: I'm glad to see that the authors included a large negative control sample group in the study and utilized it for a decontamination process later.

-Methods: I appreciate the detailed descriptions of methods.

-Methods, Collection of rectal swabs: Who collected the rectal swabs from the children? What precautions were taken to ensure that the samples would not be contaminated by the environment, skin etc?

-Methods, Mycobiota community analysis: How was the probability cutoff of 0.2 for calling contaminants chosen for the decontam process?

- Methods, Mycobiota community analysis: Were p-values for diversity analyses adjusted for multiple testing, and if so, what method was used?

-Figure 1 is readable but bad quality. For publication, a higher-quality figure is needed.

-Figure 4 seems to be very striped, making it a bit difficult to distinguish between the different genera. Figure quality should be improved for publication.

-Figure 5 has the same problems as Figures 1 and 4. Figure quality should be improved for publication.

-Supplementary figures 1-2, 4-6: Similarly to the previous comments, these figures are at times unreadable, making it difficult to assess whether the results text represents what's actually happening in the figures. The quality needs to be improved before further consideration.

-Mycobiota clusters, row 183-185: Would it be possible to list some key fungi of each cluster mentioned here? EDIT: I noticed that these have been described in a later paragraph. I'd suggest moving this up for clarity.

-Discussion, row 230-234: You seem to declare that the increased abundance of *C. orthopsilosis* here was due to the parity of the mother, however you also mention that age, quite naturally, correlates with parity. What is the reasoning in assuming that the effect was due to parity and not age of the mother?

-Discussion, general: The discussion could be improved by some additional consideration of why an antibiotic, which does not directly target fungi, could affect the mycobiota composition.

-The manuscript seems to lack a data availability statement. Authors should submit their raw sequences to a public repository and provide an identifier for the sequences. Preferably, the authors should also submit their used code in a public repository for transparency.

Reviewer #4

(Remarks to the Author)

Studying intestinal microbiota in newborns is important, but significant results require clinical explanations about the impact of microbiome changes, especially during the first two years when the immune system develops and its impairment has been associated with an increased susceptibility to respiratory and intestinal recurrent infections or skin disease like dermatitis. While the study's methodology and results are sound, the discussion lacks appeal for paediatricians and readers. For more engaging examples, see works such as "Clinical implications of maternal multikingdom transmissions and early-life microbiota" by Shuqin Zeng, Meicen Zhou, Dezhi Mu, Shaopu Wang.

Similarly, when examining the effects of climate change on intestinal microbiota modifications, it is essential to identify an explanation that has clinical relevance or elucidates the variables that contribute to these changes.

Dear Reviewers,

Thank you for reviewing our manuscript. We have attempted to address your comments as best as we could and made revisions as required. Herein we provide our point-by-point responses to each of your comments.

Reviewer #1

This is a well-written and solid manuscript. The importance of the study is well defined and leaves no question. The fungal gut microbiota is often overlooked and might have unexpected effects on particularly the infant health outcomes. This study both highlights the infant gut mycobiota development in the beginning of life, as well as the impact of azithromycin on the gut mycobiota composition.

There are only minor comments needing clarifications:

1. What was the correct number of samples in the analyses? In the Table it is written that main cohort includes n=48 in the azithromycin group and n=54 in the placebo arm. However, in the abstract the numbers of selected children are n=57 and n=70. Use in the abstract the numbers corresponding to the main cohort (and main analyses).

***Response:** The number of children initially stated in the abstract were the total number of children in each trial arm including both the main cohort and the long-term follow-up cohort. We have revised this to reflect the number of children from the main cohort only (lines 51 – 55).*

2. In the results the authors have nicely shown how different background factors associated with diversity and richness. Are these background factors included as confounding factors in the statistical tests? Or is the cohort always stratified according to these in the tests? Maybe it could be included as a confounding factors to the tests in some cases to see if the outcome is the same?

***Response:** Thank you for that question. Yes, we did factor in those variables that had effects on alpha diversity (Shannon diversity and richness) as per our multivariate analysis conducted initially. For alpha diversity comparisons between trial arms (figure 2), we did in addition to the overall comparison do a stratified analysis by season and parity as these variables showed significant effects on alpha diversity. Also, in assessing the effects of age on alpha diversity (supplementary table 3), we included season, parity, and ethnicity in our model. Beta diversity analysis was also done in a similar manner where we measured the effects of age, season, parity, and ethnicity in each trial arm (supplementary table 4).*

3. In Fig 2 the diversities are difficult to read when the background lines divide the clusters (it is difficult to see which were the actual comparisons). Consider moving the lines so they are supporting the wanted comparison.

***Response:** Thank you for that observation. We have removed the grid lines from the figures to make them easier to read.*

4. Were all children delivered vaginally?

Response: Yes, all the children included in this analysis were vaginally delivered. This has now been stated in the methods (lines 368 – 369).

5. In Table 1 recent antibiotic use was presented. Did that effect the mycobiota?

Response: Data on recent antibiotic use was only collected at year 3 and was not differentially experienced between the trial arms. Therefore, a stratified analysis of arm comparison by recent antibiotic use at year 3 was not conducted.

6. Table 1 is divided into two parts. When looking just at the table it is not clear whether the study consist of two completely different cohorts at two different timepoints or a single cohort that was followed up to 3 yrs. In Fig 1b the number of samples in control and AB treated group is outlined. From there it seems to be one cohort following the same patients. Please clarify this in the legend.

Response: All the children in this analysis were selected from the same cohort – a bacterial carriage cohort which was part of the main trial, PregnAnZI-2 (methods: lines 371 – 372). During the main trial these children were followed up to 4 months (methods: lines 380 – 382). The year three follow up was specifically for the mycobiome study which was conducted as an additional follow up for the selected children (methods: lines 372 – 374). For clarity, we have added an explanation in the foot note.

7. Discuss the limitation of using relative abundances (instead of including quantitative data as well).

Response: Thank you for highlighting this important limitation. We have now discussed this (lines 324 – 327).

8. References are not in a uniform format. Please revise.

Response: We have revised all references to the same format as per the journal's requirement.

Reviewer #2

Response: Thank you for contributing to the review of our manuscript.

Reviewer #3

Thank you for an insightful manuscript. Below, I have given an overall review of the manuscript by topic, and after that, some specific suggestions to help the authors revise the

manuscript. With this review, I recommend that this manuscript will go back to the authors for revisions before further consideration for publication.

Key results

Reviewer: The study explores the effects of antibiotics on the understudied mycobiota of the human gut. Furthermore, the trial took place in Africa, which is generally underrepresented in microbiota studies. The sample size was quite large, probably making it one of the largest longitudinal studies of its kind. The use and utilization of a large negative control sample group was impressive and left me confident that the findings largely represent a true human mycobiota rather than contamination.

Validity of the manuscript

Reviewer: I do not see immediate flaws that would prevent the publication of the manuscript outright. Some issues require addressing by the authors, especially regarding the quality of the figures, as some of the figures were so unclear it was difficult to verify the results. With sufficient revisions the manuscript could qualify for publication.

Originality and significance

Reviewer: The conclusions are original. This was a large trial with an underrepresented study population in microbiome research. I think the results should be interesting to those in the field, especially those interested in the fungal side of the microbiome, but also those interested in the effects of antibiotics on the gut microbiome.

Data & methodology

Reviewer: The quality of the data seems to be good. Methods seem appropriate for the study question. A very large negative control sample group was included, and it seemed to have been utilized well in identifying contaminant reads from the samples. The reporting of the data and methods seems sufficient for reproduction of the results.

Appropriate use of statistics and treatment of uncertainties

Reviewer: Error bars are not defined in the corresponding figure legends. This needs to be corrected with revisions. Otherwise, the statistics are generally sound. Minor clarifications are needed, as addressed in the Reviewer questions and suggestions below.

Conclusions and data interpretation

Reviewer: I think the data has been well interpreted although some of the figures need enhancing for me to be sure. The conclusions are fine, although I remain doubtful of whether the observed effect of multiparity is rather due to the age of the mother or some other undiscussed effect taking place in the study.

1. Suggested improvements/additional data

Reviewer: I want to highlight my suggestion from the Reviewer questions and suggestions regarding the data availability statement. Raw data should be uploaded to

a public repository, and if this has been done already, an identifier to the data should be provided.

Response: *We have added a data availability statement and provided links to the repository where the raw data has been deposited (lines 526 – 535).*

2. Reference list

Reviewer: Overall, the reference list seems good. Further discussion, and therefore inclusion of additional related references, could be provided regarding how antibiotics affect the mycobiota composition, considering antibiotics do not directly target fungi.

Response: *We have expanded on our discussion on the potential mechanisms through which antibiotics could exert indirect effects on the mycobiota citing relevant literature (lines 245 – 249).*

3. Clarity and context

Reviewer: The abstract is clear enough. The introduction is overall appropriate, although a bit long in my opinion. As the authors state in the limitations, the study doesn't allow straightforward conclusions on any clinical outcomes, which is why especially the beginning part in the introduction regarding maternal and neonatal deaths and severe outcomes seems a bit out of place. With this in mind, I think the focus here should rather be the mycobiota composition and possible effects on it. Speculating the effect of disrupted mycobiota on clinical outcomes would be fine in the discussion. The conclusions seem overall fine and reflect the results in this manuscript.

Response: *Thank you for pointing this out. We have revised the introduction to make it more concise and highlight the main focus of this analysis – which is to assess the effect of intrapartum azithromycin on the gut mycobiota of the child.*

4. Methods: I'm glad to see that the authors included a large negative control sample group in the study and utilized it for a decontamination process later.

Response: *Thank you.*

5. Methods: I appreciate the detailed descriptions of methods.

Response: *Thank you.*

6. Methods, Collection of rectal swabs: Who collected the rectal swabs from the children? What precautions were taken to ensure that the samples would not be contaminated by the environment, skin etc?

Response: *The samples were collected by the study nurses, who were trained on best practices for sample collection for microbiome analysis. This includes precautionary measures such as cleaning the anal area of the child with spirit before sample collection to minimize contamination with skin microbiome. Also, the nurses had appropriate gears (gloves and masks) on during the entire process and fresh gloves were used for each participant. The samples were transported in standard sample transport coolers at 4 – 8 degrees Celsius within eight hours of collection.*

7. Methods, Mycobiota community analysis: How was the probability cutoff of 0.2 for calling contaminants chosen for the decontam process?

Response: The probability cutoff was chosen after multiple tests with different cutoffs to assess the performance of the software (decontam) in identifying true contaminants in the samples. The cutoff of 2.0 showed an optimal performance, identifying most of the potential contaminants shown in the negative controls.

8. Methods, Mycobiota community analysis: Were p-values for diversity analyses adjusted for multiple testing, and if so, what method was used?

Response: Yes, we corrected for repeated testing when measuring the effect of age and other covariates on Shannon diversity. In our model, p-values were adjusted by Dunnett's test. We have now stated this in the methods (line 481).

9. Figure 1 is readable but bad quality. For publication, a higher-quality figure is needed.

Response: We have provided the high-quality pdf copy of this figure as separate file.

10. Figure 4 seems to be very striped, making it a bit difficult to distinguish between the different genera. Figure quality should be improved for publication.

Response: We have replaced this figure with one of better quality in the manuscript document and also provided the original high quality pdf version separately.

11. Figure 5 has the same problems as Figures 1 and 4. Figure quality should be improved for publication.

Response: We have replaced this figure with one of better quality.

12. Supplementary figures 1-2, 4-6: Similarly to the previous comments, these figures are at times unreadable, making it difficult to assess whether the results text represents what's actually happening in the figures. The quality needs to be improved before further consideration.

Response: We have now provided these figures in high quality format.

13. Mycobiota clusters, row 183-185: Would it be possible to list some key fungi of each cluster mentioned here? EDIT: I noticed that these have been described in a later paragraph. I'd suggest moving this up for clarity.

Response: We have moved up the taxonomic descriptions to where the clusters were initially mentioned as suggested (lines 192 – 196).

14. Discussion, row 230-234: You seem to declare that the increased abundance of *C. orthopsilosis* here was due to the parity of the mother, however you also mention that age, quite naturally, correlates with parity. What is the reasoning in assuming that the effect was due to parity and not age of the mother?

Response: Our analysis showed that maternal age inversely correlated with abundance of *C. orthopsilosis*. Given that most primiparous mothers were young mothers and this is typically the case in our study setting, we interpreted this stating that children born to primiparous mothers may carry more of this fungi than the others. However, we recognise that the effect of age here may not have direct relationship with parity and have therefore revised our statement (lines 253 – 260 and lines 346 – 349).

15. Discussion, general: The discussion could be improved by some additional consideration of why an antibiotic, which does not directly target fungi, could affect the mycobiota composition.

Response: We have expanded on our discussion in this regard citing relevant literature on how antibiotics may influence gut fungi (lines 245 – 249).

16. The manuscript seems to lack a data availability statement. Authors should submit their raw sequences to a public repository and provide an identifier for the sequences. Preferably, the authors should also submit their used code in a public repository for transparency.

Response: We have added data and code availability statements with links to the raw sequences and the codes used for the analysis (526 – 535).

Reviewer #4

1. Studying intestinal microbiota in newborns is important, but significant results require clinical explanations about the impact of microbiome changes, especially during the first two years when the immune system develops and its impairment has been associated with an increased susceptibility to respiratory and intestinal recurrent infections or skin disease like dermatitis. While the study's methodology and results are sound, the discussion lacks appeal for paediatricians and readers. For more engaging examples, see works such as "Clinical implications of maternal multikingdom transmissions and early-life microbiota" by Shuqin Zeng, Meicen Zhou, Dezhi Mu, Shaopu Wang. Similarly, when examining the effects of climate change on intestinal microbiota modifications, it is essential to identify an explanation that has clinical relevance or elucidates the variables that contribute to these changes.

Response: We have elaborated further on the influence of season on mycobiota diversity and abundance by highlighting the potential effects of specific environmental variables like temperature and humidity (line 233 – 237). We have also enhanced our discussion on the implications of some of our findings, including abundance of *Candida* and risk of infection and autoimmune diseases (lines 257 – 260 and lines 302 – 312).